

# Impacts of Air–sea Coupling on Systematic Errors in Medium-Range Winter Forecasts over the North Pacific and North Atlantic

Tien-Yiao Hsu[1], Matthew R. Mazloff[2], Sarah T. Gille[2], Hai Lin[3], K. Andrew Peterson[3], Rui Sun[2], Aneesh C. Subramanian[4], and Luca Delle Monache[1]

[1]Center for Western Weather and Water Extremes, Scripps Institution of Oceanography, University of California, San Diego, La Jolla, California, United States
[2]Scripps Institution of Oceanography, University of California, San Diego, La Jolla, California, United States
[3]Meteorological Research Division, Environment and Climate Change Canada (ECCC), Dorval, Québec, Canada
[4] Department of Atmospheric and Oceanic Sciences, University of Colorado Boulder, Boulder, Colorado, United States

**Correspondence:** Tien-Yiao Hsu (tienyiao@ucsd.edu)

**Abstract.**

The impact of air–sea coupling in North Pacific and North Atlantic medium-range forecasts during winter is assessed using 20 years (1998–2017) of hindcasts produced by the Global Ensemble Prediction System (GEPS) of Environment and Climate Change Canada (ECCC). We compare an uncoupled atmospheric model (versions 5, GEPS5) with an atmosphere–ocean coupled model (version 6, GEPS6) alongside European Centre for Medium-Range Weather Forecasts Reanalysis v5 (ERA5) as the verification dataset. We find that by the third pentad, or days 11–15, coupling weakens the Aleutian Low, the Icelandic Low, and the Atlantic Subtropical High. This produces less integrated vapor transport (IVT) over the Pacific and Atlantic Oceans, whose spatial patterns are modulated by phases of Madden–Julian oscillation (MJO). Coupling also results in colder sea surface temperature (SST) over the Kuroshio Current Extension region and produces a weaker Aleutian Low due to less upward latent heat fluxes. The weaker Aleutian Low further reinforces its weakening through a positive feedback loop. Lastly, the air–sea coupling reduces the latent heat flux bias variance by 10–20%, thus improving the IVT.

## 1 Introduction

The improvement of medium-range forecasts (5–15 days) remains critical to better prepare society for weather extremes. The time-evolving ocean states are a crucial element needed to correctly simulate strong weather variability, such as the Madden–Julian Oscillation (MJO; Madden and Julian, 1971; Wheeler and Hendon, 2004) and atmospheric rivers (ARs; Gimeno et al., 2014), that are important signals in subseasonal-to-seasonal (S2S) precipitation forecasts (Subramanian et al., 2019). Mid-latitude cyclones can cause strong sea surface temperature (SST) perturbations (Hsu et al., 2024) approximately 10 days after passage (Kobashi et al., 2019) and feed back to the storm tracks (Booth et al., 2012). Major weather agencies have adopted high-resolution (less than 50 km) coupled systems for medium-range forecasts and have shown detectable improvement in



forecast skill through the use of coupled models (Brassington et al., 2015). The benefit often comes from the tropics, where cloud convection is an important source of available potential energy and is sensitive to SST.

Since air–sea fluxes are modulated by near-surface wind speed, two-way air–sea coupling measurably improves tropical cyclone forecasts. The SST cooling induced by wind-driven ocean mixed-layer deepening and Ekman upwelling can feed back in a few days to reduce storm intensity (Rainaud et al., 2017; Smith et al., 2018; Sun et al., 2022; Polichtchouk et al., 2025).

Similarly, coupling is also known to have a positive impact on MJO prediction (DeMott et al., 2015; Savarin and Chen, 2022) because SST cooling due to wind anomalies and the diurnal variation of mixed-layer depth can modulate MJO propagation speed and intensity. The ability to predict the MJO is particularly important because it is known to remotely modulate the North Atlantic Oscillation (NAO) (Cassou, 2008; Lin et al., 2009; Scaife et al., 2017) and to influence global temperature and precipitation on a subseasonal timescale (Stan et al., 2017).

Coupled models have advanced to grid sizes of less than a degree, leading to new understandings of air–sea coupling. In particular, there is a growing awareness of the role of ocean-eddy-scale air–sea interactions in high-resolution simulations where the SST gradients can effectively modify near-surface atmospheric curl, divergence, and therefore heat fluxes (Small et al., 2008; Bishop et al., 2017; Liu et al., 2021; Seo et al., 2023; Renault et al., 2024). However, over western boundary current extensions coupled models do not necessarily predict the SST within eddies better than persistence (Vellinga et al.,

2020), contributing to systematic errors in medium-range forecasts.

In this study, we focus on the impact of air–sea coupling in the North Pacific and North Atlantic during the winter. We take advantage of archived hindcasts of the Global Ensemble Prediction System (GEPS) versions 5 (GEPS5) and 6 (GEPS6) of the Environment Climate Change Canada (ECCC) provided as part of the subseasonal-to-seasonal (S2S) project (Vitart et al., 2017). Because GEPS5 is run with prescribed sea surface conditions and GEPS6 with interactive sea surface conditions from

a coupled ocean model, their contrasts reveal the impact of air–sea coupling. The global models can see the full impact of air–sea coupling, a potential advantage over regional models, where the coupling effects are constantly removed as the air keeps flowing in from the prescribed lateral boundary conditions. We use integrated vapor transport (IVT; Rutz et al., 2014) as a proxy to assess weather extremes due to its connection with atmospheric rivers (ARs; Zhu and Newell, 1994; Gimeno et al., 2014; Rutz et al., 2014; Guan and Waliser, 2015; Pasquier et al., 2019; Waliser and Guan, 2017). Overall, we find three

main points. First, the coupling weakens the Aleutian Low, the Icelandic Low, and the Atlantic Subtropical High, subsequently resulting in a weaker IVT whose spatial pattern is influenced by the MJO. Second, the coupling simulates a colder SST over the Kuroshio Current extension, generating a weaker Aleutian Low, which further weakens itself through a positive feedback loop. Third, the air–sea coupling reduces the latent heat flux bias variance by 10–20% and improves the IVT forecast over the Kuroshio Current Extension.

In Section 2, we introduce our datasets and methodology. Section 3 presents and discusses our results. In Section 4, we draw conclusions.



## 2 Dataset and Methods

### 2.1 GEPS

GEPS5 uses the Global Environmental Multiscale (GEM) atmospheric model (Côté et al., 1998b, a). GEPS5 has 45 vertical
levels using log-pressure vertical coordinate (Girard et al., 2014), and uses the Ying–Yang grid with a horizontal resolution of
39 km (Qaddouri and Lee, 2011). As for the ocean boundary condition, GEPS5 uses the persistent anomaly method: on top of
the climatological seasonal cycle, the 30-day average SST anomaly preceding the initial date is added and persists throughout
the integration (Lin et al., 2016). The sea ice cover is adjusted according to local SST so that the resulting sea ice cover and SST
are consistent (Gagnon et al., 2014). The initial conditions are obtained using an Ensemble Kalman-filter (EnKF; Houtekamer
et al., 2009, 2014), with a digital filter (Fillion et al., 1995) and incremental analysis updates (Bloom et al., 1996) to reduce the
shock during data assimilation (Deng et al., 2018).

GEPS6 is built on top of GEPS5 by replacing the simple statistical SST and sea ice model with a dynamical ocean and
sea ice model. The ocean model is the Nucleus for European Modelling of the Ocean (NEMO) version 3.6 (Madec, 2008).
NEMO uses $z$-level vertical coordinates, with hydrostatic, Boussinesq approximations, and a linear free surface. This version
has a horizontal resolution of 0.25° ORCA grid (Bernard et al., 2006, a global tripolar grid configured to remove singularity of
poles of a sphere) and 50 levels increasing from 1 m at the surface to 500 m at the deepest level. The sea ice model is the Los
Alamos multi-category Community Ice Model version 4 (CICE4; Hunke, 2001; Lipscomb et al., 2007; Hunke et al., 2015). The
initial conditions are obtained using the EnKF, with European Centre for Medium-Range Weather Forecasts hybrid (ECMWF-
hybrid) gain applied to recenter ensemble members around the means of EnKF analysis and 4DEnVar analysis (Penny, 2014;
Houtekamer et al., 2019). The 4DEnVar is a 4-dimensional variational data assimilation using the Global Deterministic Pre-
diction System (Buehner et al., 2015; Lin et al., 2019).

For more detailed documentation, see Peterson et al. (2022) and Smith et al. (2018). Ensemble methods are described by
Deng et al. (2018) for GEPS5 and Lin et al. (2019) for GEPS6.

### 2.2 Hindcast Data

The S2S project provides up to 60 lead days hindcasts (Vitart et al., 2008, 2017) from 13 different meteorological agencies.
ECCC has contributed hindcast data from GEPS5 and GEPS6 from 1998–2017.

The hindcasts are produced on a weekly basis for GEPS5 and GEPS6. For each hindcast date, hindcasts corresponding to
the same date were generated for 20 years 1998–2017. Each hindcast has a lead time of 32 days with 4 ensemble members. For
GEPS6, the hindcast is generated such that it has twice as many start dates as the GEPS5 hindcast, as documented in Tables S1
and S2. We subsample the GEPS6 hindcast by choosing the closest start date (underlined in Table S2) to GEPS5. This strategy
minimizes the impact of the start time difference and ensures that the GEPS6 subset has the same amount of data as GEPS5.
In our focus months, December, January, and February, the start dates of GEPS5 are exactly one day later than GEPS6.

As our verification dataset, we use European Centre for Medium-Range Weather Forecasts (ECMWF) Reanalysis v5 (ERA5;
Hersbach et al., 2020). In the Pacific and Atlantic Oceans, it can well capture offshore diurnal SST cycles under various wind



conditions (Yao et al., 2021). Over Europe, the wind variability is skillfully predicted (Molina et al., 2021; Chen et al., 2024). Over North America, Chen et al. (2024) shows that ERA5 has skills over wind and precipitation associated with extra-tropical cyclones, with a tendency to underestimate high winds and overestimate low winds. A study over the Red Sea shows ERA5 is challenged by land–sea induced local dynamics (Alkhalidi et al., 2025).

## 2.3 Assessment of air–sea coupling

Ideally, the effect of air–sea coupling can be revealed by taking the difference between coupled and uncoupled hindcasts. However, GEPS5 and GEPS6 do not share common start times. Therefore, we reference both fields to ERA5 by computing the difference between GEPS and ERA5 data and then sorting the runs by start month. We first define the pentad bias

$$\beta_{\mathrm{pdt},X}\left(\boldsymbol{r},t_s,p,\gamma\right) = \left(\Delta w\right)^{-1} \int_{t_l=(p-1)\,\Delta w}^{p\,\Delta w} X_{\mathrm{pdt,hcst}}\left(\boldsymbol{r},t_s,t_l,\gamma\right) - X_{\mathrm{ref}}\left(\boldsymbol{r},t_s+t_l\right)\,\mathrm{d}t_l, \tag{1}$$

where $\beta_{\mathrm{pdt},X}$ is the bias of the hindcast product "pdt" of the variable $X$ at location $\boldsymbol{r}$, $t_s$ is the start time, $t_l$ is the lead time, $p$ is the lead pentad (starting from 1), $\gamma$ is the ensemble member of a total $N_\gamma$ members, and $\Delta w = 5$ day is the size of the pentad. The subscript "hcst" denotes the hindcast, "ref" denotes the reference dataset that is used to verify the hindcast, i.e., ERA5 in this paper. With the $\langle\cdot\rangle$ being the spatial averaging over a region $S$, we can separate the bias into a spatial mean $\langle\beta_{\mathrm{pdt},X}\rangle$ and an anomaly $\beta'_{\mathrm{pdt},X} = \beta_{\mathrm{pdt},X} - \langle\beta_{\mathrm{pdt},X}\rangle$. The averaged bias variance can then be written as the sum of mean and patterned variances. That is,

$$\left\langle\beta_{\mathrm{pdt},X}^2\right\rangle = \overbrace{\left\langle\beta_{\mathrm{pdt},X}\right\rangle^2}^{\mathrm{mean}} + \overbrace{\left\langle\beta'^2_{\mathrm{pdt},X}\right\rangle}^{\mathrm{patterned}}. \tag{2}$$

Later in the text for the sake of simplicity, we define bias variance $\epsilon_{\mathrm{pdt},X} = \left\langle\beta_{\mathrm{pdt},X}^2\right\rangle$, mean bias variance $\bar{\epsilon}_{\mathrm{pdt},X} = \left\langle\beta_{\mathrm{pdt},X}\right\rangle^2$, and patterned bias variance $\tilde{\epsilon}_{\mathrm{pdt},X} = \left\langle\beta'^2_{\mathrm{pdt},X}\right\rangle$, with the decomposition as $\epsilon_{\mathrm{pdt},X} = \bar{\epsilon}_{\mathrm{pdt},X} + \tilde{\epsilon}_{\mathrm{pdt},X}$.

The bias and its variance decomposition of variable $X$ of a product pdt grouped by start time set $\phi$ is

$$B_{\mathrm{pdt},X}\left(\boldsymbol{r},\phi,p\right) = \left\{\beta_{\mathrm{pdt},X}\left(\boldsymbol{r},t_s,p,\gamma\right) \mid t_s \in \phi, \gamma = 1,\ldots,N_\gamma\right\}, \tag{3a}$$

$$E_{\mathrm{pdt},X}\left(S,\phi,p\right) = \left\{\epsilon_{\mathrm{pdt},X}\left(S,t_s,p,\gamma\right) \mid t_s \in \phi, \gamma = 1,\ldots,N_\gamma\right\}, \tag{3b}$$

$$\overline{E}_{\mathrm{pdt},X}\left(S,\phi,p\right) = \left\{\bar{\epsilon}_{\mathrm{pdt},X}\left(S,t_s,p,\gamma\right) \mid t_s \in \phi, \gamma = 1,\ldots,N_\gamma\right\}, \tag{3c}$$

$$\tilde{E}_{\mathrm{pdt},X}\left(S,\phi,p\right) = \left\{\tilde{\epsilon}_{\mathrm{pdt},X}\left(S,t_s,p,\gamma\right) \mid t_s \in \phi, \gamma = 1,\ldots,N_\gamma\right\}, \tag{3d}$$

To test the significance, the degrees of freedom are counted by making the following two assumptions: (a) output from different start times or different ensembles is independent, and (b) the output within the same pentad is not independent. In both GEPS5 and GEPS6, during 1998–2017 there are 4 start times in January with 4 ensemble members. Therefore, for each pentad there are $20 \times 4 \times 4 = 320$ degrees of freedom.

To assess the impact of air–sea coupling as a function of space, we define bias change

$$\Delta B_X\left(\boldsymbol{r},\phi,p\right) = \mu\left[B_{\mathrm{GEPS6},X}\left(\boldsymbol{r},\phi,p\right)\right] - \mu\left[B_{\mathrm{GEPS5},X}\left(\boldsymbol{r},\phi,p\right)\right], \tag{4a}$$





where $\mu$ is the averaging operator over a given set. A significance test is performed with the above-mentioned degrees of
freedom.

## 2.4  Separation of MJO impact

To evaluate MJO impact, we define three start time groups using the outgoing-longwave-radiation (OLR)-based MJO index
(OMI; Kiladis et al., 2014), a two-dimensional vector whose values are normalized principal components. When the magnitude
of OMI is less than 1, the MJO is classified as inactive. When the magnitude of OMI is larger than 1, the MJO is active, and
the MJO phases 1–8 are defined according to the phase angle of OMI. MJO phase contains spatial information of MJO: during
MJO phases 1–4, the MJO convection center resides over the Indian Ocean. During MJO phases 5–8, the center is over the
Maritime continent and tropical Pacific. The MJO start time groups are defined as

$$\phi_{\text{NonMJO}} = \{t \mid t \in \phi_{\text{DJF}} \text{ and the MJO is inactive more than half of the time in the next 15 days.}\} \tag{5a}$$

$$\phi_{\text{P1234}} = \{t \mid t \in \phi_{\text{DJF}}, \text{ the MJO is in phases 1–4 more than half of the time in the next 15 days.}\} \tag{5b}$$

$$\phi_{\text{P5678}} = \{t \mid t \in \phi_{\text{DJF}}, \text{ the MJO is in phases 5–8 more than half of the time in the next 15 days.}\} \tag{5c}$$

where $\phi_{\text{DJF}}$ is the set of all start times during December–January–February. The remaining start times are ambiguous, meaning
that either the MJO is neither consistently inactive nor active, or the phase of MJO cannot be classed in either P1234 or P5678.
Out of 1805 days of DJF during 1998–2017, there are 455 days of NonMJO, 331 days of P1234, 321 days of P5678, and 698
days that are ambiguous. (See Figure S1 for histogram.)

# 3  Results

## 3.1  Systematic Impact of Air–Sea Coupling On SST and Circulation

We compute the bias change $\Delta B$, i.e., the difference between GEPS6 and GEPS5, of SST, 850 hPa geopotential height $Z_{850}$,
and 500 hPa geopotential height $Z_{500}$ of pentads 1–3 and present them in Figure 1. The black boxes in Figure 1c define
the Kuroshio Current Extension (150°E–130°W, 30°–50°N) and the Gulf Stream (75°–15°W, 35°–55°N) regions, and the
magenta boxes in Figures 1f and 1i define the Aleutian Low (140°E–130°W, 40°–60°N), Icelandic Low (30°W–20°E, 55°–
70°N), and Atlantic Subtropical High (60°W–20°E, 20°–50°N) regions.

Throughout all pentads, the pattern of SST bias is consistent, with increased magnitude in later time (Figures 1a–c). Over
the North Pacific, the SST is colder in GEPS6, with the alternating signs around coastal Japan signifying the error in simulating
the KC. We see a similar cold bias in the North Atlantic, but with a strong northward shift of the Gulf Stream along 45°N. The
impact of this shift extends northward all the way to the edge of Arctic sea ice.

As for the circulation, by pentad 3, $Z_{850}$ shows a weakening in the Aleutian Low, the Icelandic Low, and the Atlantic
Subtropical High (Figures 1f and 1i). The weakening of the Aleutian Low starts in pentad 1 and grows until pentad 3. The
weakening of Icelandic Low starts remotely in the middle of Eurasia as a positive anomaly in pentad 1. Then, the anomaly





**Figure 1.** Bias changes $\Delta B$ of atmosphere quantities computed from Global Ensemble Forecast System (GEPS) version 5 (GEPS5) to GEPS version 6 (GEPS6) during December–January–February of the first three pentads in hindcast years 1998–2017. (a)–(c) $\Delta B$ of the sea surface temperature (SST) of pentad = 1, 2, and 3. (d)–(f) Same as a–c but for 500 hPa geopotential height $Z_{850}$. (g)–(i) Same as (a)–(c) but for 500 hPa geopotential height $Z_{500}$. The hatched area passes the significance test of a $p$-value of 0.1. The black boxes in panel c define the Kuroshio Current Extension (150°E–130°W, 30°–50°N) and Gulf Stream (75°–15°W, 35°–55°N) regions, and the magenta boxes in panels f and i define the Aleutian Low (140°E–130°W, 40°–60°N), Icelandic Low (30°W–20°E, 55°–70°N), and Atlantic Subtropical High (60°W–20°E, 20°–50°N) regions.

grows and moves westward into the location of the Icelandic Low. Meanwhile, the weakening of the Atlantic Subtropical
High emerges in pentad 2, and its magnitude becomes comparable to the weakening of the Icelandic Low. Together with the





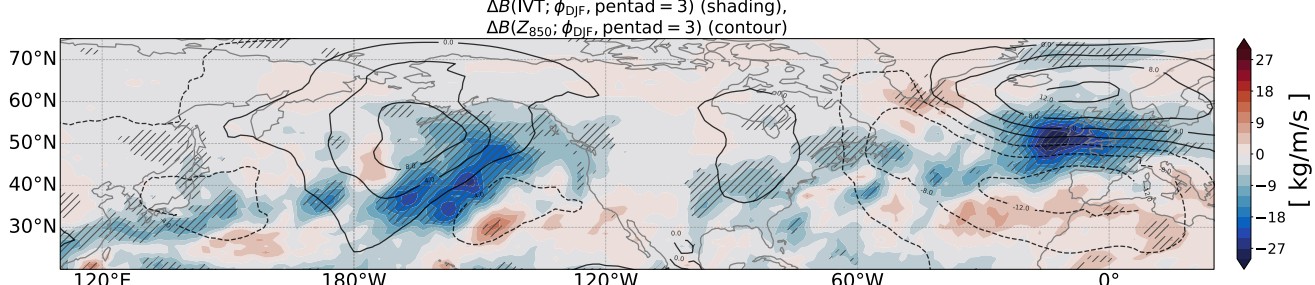

**Figure 2.** Bias changes $\Delta B$ of the integrated vapor transport (IVT, shading) and 850 hPa geopotential height $Z_{850}$ (contours, spacing is 2 meters, and contours with negative values are dashed)) computed from Global Ensemble Forecast System (GEPS) version 5 (GEPS5) to GEPS version 6 (GEPS6) during December–January–February of the first three pentads in hindcast years 1998–2017. The hatched area means the IVT anomalies pass the significance test of a $p$-value of 0.1.

weakened Icelandic Low, the westerly winds in between ($45°$–$60°$N) decelerate. This deceleration is not restricted to the lower atmosphere but extends to $Z_{500}$ (Figures 1g–i).

The Aleutian Low weakening appears to be linked to a similar Icelandic Low weakening through a Rossby wave train (Hoskins and Karoly, 1981; Karoly, 1983). As shown in Figures 1f and 1i, $Z_{850}$ and $Z_{500}$ reveal an alternating pattern of high

and low centers, extending from the Aleutian Low through the Arctic to the Icelandic Low. The wave train pattern is more significant in $Z_{850}$ than $Z_{500}$, suggesting that the impact originates from the lower atmosphere.

The large northward shift of the Gulf Stream directly forces $Z_{850}$ and $Z_{500}$. Figures 1d and 1g show that there are positive geopotential anomalies at $45°$N, $75°$W. The anomalies persist throughout pentads 1–2.

### 3.2  Systematic Impact of Air–Sea Coupling on Integrated Vapor Transport (IVT)

Here, we define IVT $= \left| g^{-1} \int_{200\text{hPa}}^{1000\text{hPa}} q \, \boldsymbol{v} \, \mathrm{d}p \right|$. Figure 2 shows the bias change $\Delta B$ of the IVT (shading) and $Z_{850}$ (contours) for pentad 3. Over the North Pacific, the IVT is reduced along the southeastern side of the weakened Aleutian Low towards the Gulf of Alaska. Over the North Atlantic, the reduced IVT lies between the weakened Icelandic Low and the Atlantic Subtropical High, with a more zonal orientation toward western Europe.

The shape of IVT bias depends on the MJO. Figures 3a–c show the composite bias changes of IVT (shading) and $Z_{850}$

(contour) grouped by MJO-inactive, MJO phase 1–4, and MJO phase 5–8 as defined in Section 2.4. The weakened Aleutian Low remains in the middle of the North Pacific, such that the weakened Pacific IVT is consistently oriented southwest– northeast. In contrast, the Icelandic Low and Atlantic Subtropical High weakening is spatially more variable, such that the Atlantic IVT pattern is less consistent among MJO groups.





### 3.3 What Causes the Aleutian Low Weakening?

In this section, the goal is to understand the physical mechanisms that cause changes in the Aleutian Low. From Figures 3a–c, we know that the weakening bias is consistently simulated in GEPS6 relative to GEPS5, regardless of whether its shape is modulated by the MJO phase. This is an indication that local air–sea coupling is an important driver for the Aleutian Low weakening.

To remove the MJO influence, we examine the response of $Z_{850}$ and latent heat fluxes $H_{\mathrm{lat}}$ composited with non-MJO
groups. In the absence of MJO, the Aleutian Low weakens in GEPS6 relative to GEPS5 during pentads 1–2 (contours in Figures 4a–d). The SST bias change between $30°$–$60°$N is $-0.2$ K in the first pentad, indicating that the coupling results in a colder SST (Figure 4a shading). This results in a reduction in $H_{\mathrm{lat}}$ (Figure 4c shading), causing a weaker cyclogenesis such that the $Z_{850}$ is positively biased.

The Aleutian Low experiences a positive feedback loop where its initial weakening leads to further intensification of this
weakening, primarily through interactions between circulation and $H_{\mathrm{lat}}$. In particular, notice that the magnitude of the reduction in $H_{\mathrm{lat}}$ is larger at the southern flank of the $Z_{850}$ anomaly because its anomalous circulation blows against the mean westerlies (Figure 4c). As previously demonstrated, the reduction of $H_{\mathrm{lat}}$ leads to a weaker Aleutian Low, resulting in stronger anomalous circulation in the next pentad (Figure 4d). This mechanism is a positive feedback.

### 3.4 Two-way Coupling Improves Latent Heat Fluxes and IVT Prediction

We use error variances as functions of lead pentads over the Kuroshio Current Extension and Gulf Stream regions to assess how coupling benefits prediction, as shown in Figures 5a–j. In the Kuroshio Current Extension region, GEPS6 performs better than GEPS5 in terms of the mean SST variance, but has poorer performance in patterned SST variance (Figure 5a). The patterned SST variance hindcast gradually reaches the mean ERA5 SST variance $0.60 \pm 0.16$ K (zonally detrended and area weighted variance, years 1998–2017 DJF). In GEPS6, the SST error is largely due to the hindcast initialization strategy of initializing
with a monthly average Ocean Reanalysis Pilot 5 (ORAP5; Zuo et al., 2017) product. These errors are not representative of the impact of coupling, nor of SST error in the hindcast, which is shown to be improved in GEPS6 compared to GEPS5 (Lin et al., 2019, Figure 24). The use of a monthly products for initializing sea ice in the hindcast similarly leads to a large SST error signal along the Arctic sea ice.

Interestingly, in the Kuroshio Current Extension region the latent heat flux $H_{\mathrm{lat}}$, GEPS6 outperforms GEPS5 in both mean
and patterned variances (Figure 5b), and the error variance $E$ of GEPS6 is 10–20% smaller than that of GEPS5. Because GEPS6 produces a less accurate SST but a better latent heat flux, this contrast highlights the importance of two-way coupling in correctly predicting air–sea fluxes, especially those associated with extra-tropical cyclones (Kobashi et al., 2019; Hsu et al., 2024). The improvement of $H_{\mathrm{lat}}$ subsequently leads to a better integrated water vapor (IWV, defined as $\mathrm{IWV} = g^{-1} \int_{200\mathrm{hPa}}^{1000\mathrm{hPa}} q\,\mathrm{d}p$) and therefore better IVT hindcast in GEPS6 (Figures 5c and d). In contrast, while air–sea coupling modifies $Z_{850}$ (Figure
1a–c), it does not produce a better $Z_{850}$ hindcast (Figure 5e).





**Figure 3.** Bias changes $\Delta B$ of the integrated water vapor (IVT, shading) and 850 hPa geopotential height $Z_{850}$ (contours, spacing is 2 meters, and contours with negative values are dashed) from Global Ensemble Forecast System (GEPS) version 5 (GEPS5) to GEPS version 6 (GEPS6) of the third pentad $p = 3$ in different start time groups in hindcast years 1998–2017 during December–January–February. (a) Non-MJO group. (b) P1234 group. (c) P5678 group. The hatched area means the IVT bias change passes the significance test of a $p$-value of 0.1.

In the Gulf Stream region, GEPS6 simulates a better mean variance of SST bias in the first three pentads. On the contrary, because GEPS6 simulates a northward-shifted Gulf Stream, there is a strong patterned variance of SST bias (Figure 5f). This signal propagates into patterned variance of $H_{\text{lat}}$ bias (Figure 5g), resulting in little or no improvement in IWV and IVT (Figures 5h–i). Moreover, similar to the Kuroshio Current Extension region, we do not see a notable difference in $Z_{850}$ (Figure 5j).






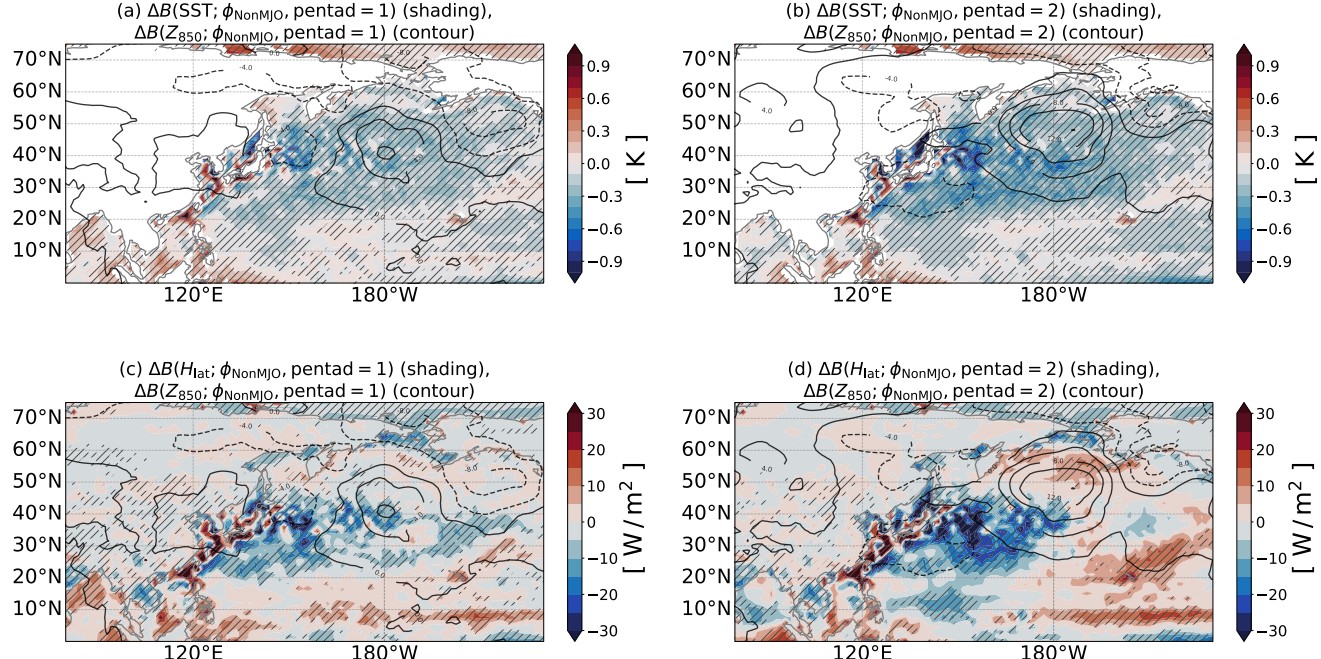

**Figure 4.** Bias change $\Delta B$ of the sea surface temperature (SST), upward latent heat flux ($H_\text{lat}$), and the 850 hPa geopotential height $Z_{850}$ from Global Ensemble Forecast System (GEPS) version 5 (GEPS5) to GEPS version 6 (GEPS6) during Madden–Julian–Oscillation (MJO) inactive start time ($\phi_\text{NonMJO}$) of the first two $p = 1, 2$ in hindcast years 1998–2017. (a) The shading is the $\Delta B_\text{SST}(\phi_\text{NonMJO}, p = 1)$. The contours are the $\Delta B_{Z_{850}}(\phi_\text{NonMJO}, p = 1)$. The hatched areas are the location where $\Delta B_{H_\text{SST}}$ passes the significance test of a $p$-value of 0.15. (b) Same as (a) but for $p = 2$. (c) Same as (a) but the shading and dotted-hatch are for the $\Delta B_{H_\text{lat}}(\phi_\text{NonMJO}, p = 1)$. (d) Same as (c) but for $p = 2$.

## 4 Discussion: The Role of Kuroshio Current Extension and the Gulf Stream

The Kuroshio Current Extension is an eddy-rich area where mesoscale (200 km and less) SST fronts modify the near-surface atmospheric convergence, curl, and thus air–sea fluxes in the marine atmosphere boundary layer (MABL) (Bishop et al., 2017; Seo et al., 2023; Renault et al., 2024). Our results show that the bias of SST over Kuroshio Current Extension leads to an

Aleutian Low positive feedback response, implying that better Aleutian Low prediction can be achieved through optimizing the initialization. This will lead to better forecasts of the North Pacific jet and IVT, both of which are indicators for AR activities (Winters, 2021; Higgins et al., 2024).

The role of the Gulf Stream is less clear. While its persistent impact on $Z_{850}$ is visible (Figure 3d–f), and the major atmospheric response over the Atlantic Ocean is immediately downstream of the Gulf Stream, the response does not emerge until

the second pentad. Therefore, while the shift of the Gulf Stream may lead to the dipole patterns of $Z_{850}$ and $Z_{500}$ over the Atlantic Ocean, this signal is mixed with remote impacts. Future numerical studies are needed to gain a deeper understanding.



## 5 Conclusion

This study, using 20 years of hindcast data from ECCC's GEPS5 and GEPS6 alongside ERA5 reanalysis, demonstrates that air–sea coupling can impact medium-range wintertime forecasts over the North Pacific and North Atlantic.

The analysis of hindcast bias shows that the air–sea coupling results in a weaker Aleutian Low, Icelandic Low, and Atlantic Subtropical High within 15 days, leading to a weaker IVT over the northeastern Pacific and Atlantic. We also notice a possible teleconnection from the Aleutian Low through the Arctic into Icelandic Low via a Rossby wave train. Furthermore, the MJO phase can influence the resulting spatial distribution of IVT difference, suggesting its importance in tropical–extratropical interactions.

We investigated the cause of the weaker Aleutian Low due to air–sea coupling and found a physical mechanism. The coupling simulates a colder SST centered on the Kuroshio Current Extension, which reduces the latent heat flux. This leads to weaker cyclogenesis and thus a weaker Aleutian Low. The anomalous circulation that blows against the westerlies over the Kuroshio Current Extension further reduces the latent fluxes, creating a positive feedback loop that reinforces the initial bias.

    When evaluating the bias variance, we find that the coupled model produces a slight degradation in SST hindcast, but a

significant 10–20% less latent heat flux bias variance over the Kuroshio Current Extension compared to the uncoupled model. The improvement in latent heat flux explains the better IWV and thus the IVT hindcast. The IVT improvement is also more significant when the MJO is active. In the Gulf Stream, the northward shift bias is too strong such that the latent heat fluxes, and thus IWV and IVT, are not improved.

    In conclusion, we find that optimizing the air–sea fluxes parameterization can lead to tangible improvements in medium-

range forecasts in the Pacific and potentially the Atlantic. Moreover, the accurate representation of the Gulf Stream may improve the Atlantic circulation bias in the atmosphere.

    Finally, this research highlights two potential future directions. First, regional simulations over the North Atlantic can be performed to isolate the influence of the Atlantic from the Pacific (Cassou, 2008). Second, there is a need for more physical understanding of how two-way coupling produces better air–sea fluxes. This potentially can mitigate the SST error along the

Kuroshio Current Extension and the Gulf Stream that can tangibly force the atmosphere through modifying air–sea fluxes (Seo et al., 2023).

*Code and data availability.* The code used to generate the figures in this study has been deposited in https://github.com/meteorologytoday/paperfigures-airsea-cpl-ECCC. The data used to generate figures in this study have been deposited in Zenodo (https://doi.org/10.5281/zenodo.16938865). The GEPS5 and GEPS6 output can be obtained from ECMWF S2S Data Repository (https://apps.ecmwf.int/datasets/data/s2s-realtime-daily-

averaged-cwao/)

*Author contributions.* Conceptualization, T.-Y.H.; Funding and resource acquisition, all authors; Investigation, T.-Y.H.; Project administration, T.-Y.H.; Visualization, T.-Y.H.; Writing–original draft, T.-Y.H.; Writing–review and editing, all authors.



*Competing interests.* Authors confirm that there are no competing interests present.

*Acknowledgements.* The authors would like to acknowledge support from CW3E through the California Department of Water Resources
funded Atmospheric River Program Phase IV (contract number 4600014942). Authors acknowledge NOAA grants NA21OAR4310257 and
NA22OAR4310597. M.R.M. acknowledges support from NASA award 80NSSC23K0979. S.T.G. acknowledges support from NASA award
80GSFC24CA067. A.C.S. acknowledges support from NOAA NA22OAR4310599, ONR ASTraL research initiative N00014-23-1-2092 and
NASA 21-OSST21-0026. This work is based on S2S data. S2S is a joint initiative of the World Weather Research Programme (WWRP) and
the World Climate Research Programme (WCRP). The original S2S database is hosted at ECMWF as an extension of the TIGGE database.





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





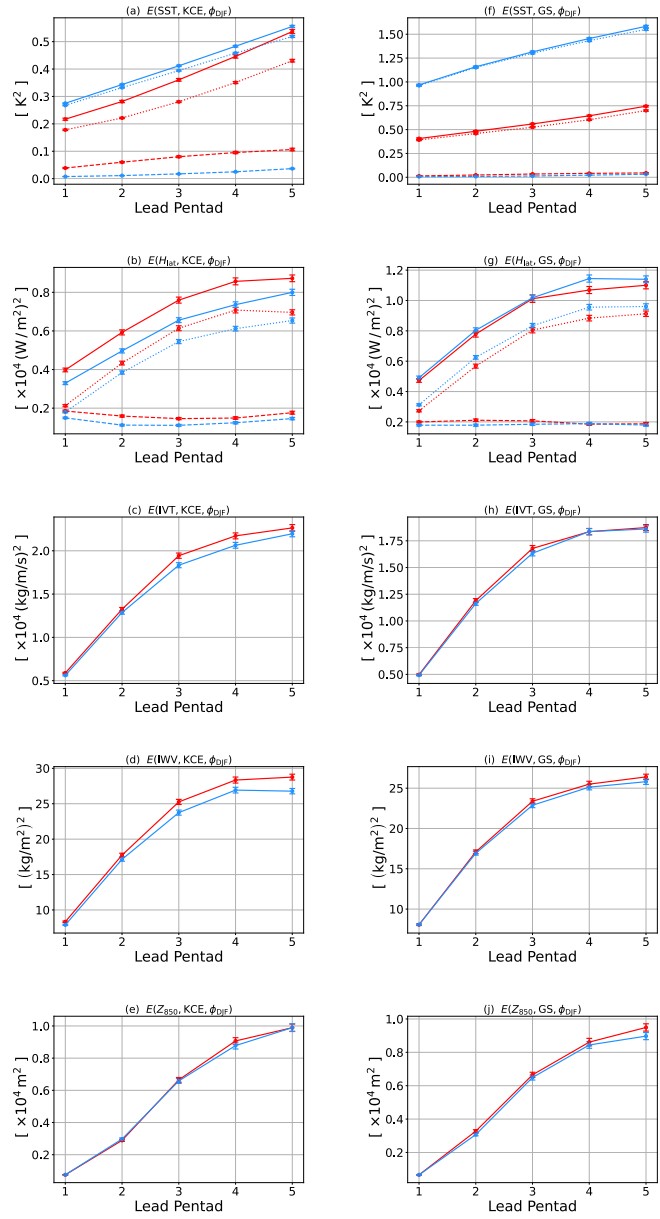

**Figure 5.** Bias variance $E$ analysis of quantities as a function of pentads 1–6 computed from Global Ensemble Prediction System (GEPS) version 5 (GEPS5, red) to GEPS version 6 (GEPS6, blue) during December–January–February of the first three pentads in hindcast years 1998–2017. (a, f) sea surface temperature (SST). (b, g) Latent heat flux ($H_{\mathrm{lat}}$. (c, h) Integrated vapor transport (IVT). (d, i) Integrated water vapor (IWV). (e, j) 850 hPa geopotential height $Z_{850}$. For SST (a and f) and $H_{\mathrm{lat}}$ (b and g), the decomposition of $E$ into mean ($\overline{E}$, dashed) and patterned ($\tilde{E}$, dotted) variances are added. Panels a–e are for the Kuroshio Current Extension (KCE) region, and f–j are for Gulf Stream (GS) region. The whiskers represent the standard error.





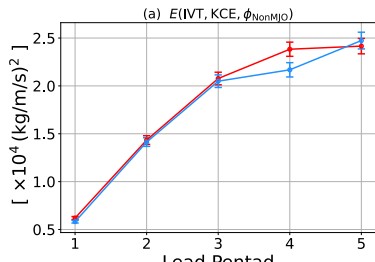
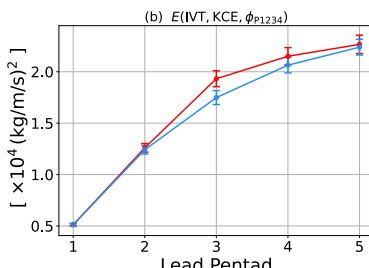
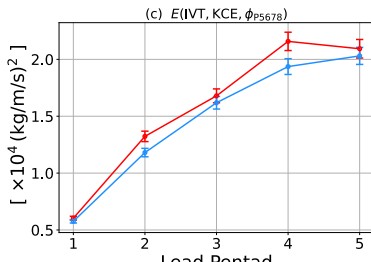

**Figure 6.** Bias variance $E$ analysis of integrated vapor transport (IVT) as a function of pentads 1–6 computed from Global Ensemble Prediction System (GEPS) version 5 (GEPS5, red) to GEPS version 6 (GEPS6, blue) using different start time groups in hindcast years 1998–2017 for Kuroshio Current Extension region. (a) Non-MJO group. (b) P1234 group. (c) P5678 group. The whiskers represent the standard error.