# Peer review of "Impacts of Air—sea Coupling on Systematic Errors in Medium-Range Winter Forecasts over the North Pacific and North Atlantic"

_EGUsphere, 2025_

## Referee Comment (RC2)

**GENERAL COMMENTS**

This paper compares ensemble forecasts of the GEPS6, which is dynamically coupled to NEMO, with hindcasts of GEPS5, which uses persisted SST anomalies. The authors find changes in the mean and variance between the two forecasts, both in the ocean and the atmosphere. The changes are discussed in terms of the effect of air-sea coupling.

While there is some interesting analysis here, I think the paper as it currently stands has several major problems. Some of this relates to the framing of the paper (the title makes it out to be about air-sea coupling but is actually about the broader effect of using a dynamical ocean model), and some of it relates to a potentially serious confounding effect (the initialisation of GEPS6 hindcasts is apparently very different from GEPS5 hindcasts). There are also many missing references to past literature.

I flesh out these and other issues in my comments below. Major revisions will be needed to address them. I look forward to reading a revised version.

Best wishes, Kristian Strommen

**MAJOR COMMENTS**

1. The paper is framed in terms of the effect of air-sea coupling, but the comparison of the two forecasts is much better thought of as examining the impact of using a dynamically coupled ocean model. The distinction is important: the difference is not just that there is an exchange of information now between ocean and atmosphere, but that the dynamic ocean introduces its own unique SST biases (which it will have even when run without an atmosphere, since it's not a perfect model). Some of the changes documented in the paper seem to be about changes to the biases and not really about the two-way coupling. For example, the change in the Gulf Stream is consistent with the fact that NEMO at ¼ degree resolution does not simulate a Gulf Stream that separates from the continent correctly. This does not have anything to do with coupling (it happens also in ocean-only simulations), but is related to model resolution and bathymetry.

Studies that aim to really isolate coupling often deal with this by looking at things like lead-lag correlations between SSTs and wind-stress or fluxes, since correlations ignore magnitude and thus are insensitive (at least a priori) to model biases. For an example, see e.g. <a href="https://doi.org/10.1002/2016GL070559">https://doi.org/10.1002/2016GL070559</a>.

I think unless you want to almost completely redo the paper to follow similar methods, you need to reframe the paper to be much more specifically about the impact of using a coupled dynamical ocean model in your forecast. However, at this

point it's clear the results depend sensitively on the exact model, since this determines the model biases. Thus, I think the authors should rephrase everything to be very specifically about the comparison between GEPS6 and GEPS5. This includes mentioning GEPS somewhere in the title. Air-sea coupling should not be mentioned in the title unless considerable additional analysis along the lines of the Roberts et al. paper (or similar) is added.

- 2. Following on from the above, the relationship between this paper and the tech report of Lin et al. (2019) is highly unclear. It seems Lin et al. already look at the impact of forecast skill, but this is only mentioned all of a sudden in the middle of the discussion of your results. This surprised me given that Lin is a co-author here as well! You need to mention Lin et al. (2019) in the Introduction, and clearly discuss what their results are and how your analysis and results differ or complement theirs.
- 3. Halfway through the paper you write the following: "These errors are not representative of the impact of coupling, nor of SST error in the hindcast, which is shown to be improved in GEPS6 compared to GEPS5 (Lin et al., 2019, Figure 24).". This is startling to say the least. It sounds like you're saying that the comparison between GEPS6 and GEPS5 you are making here is not telling you anything about either coupling nor SST biases because the initialisation of GEPS6 is so different from that of GEPS5. Doesn't this compromise every single result of this paper? Aren't you trying to exactly assess the impact of coupling or SST errors in the hindcasts? Can you please clarify the exact differences in the initialization between GEPS6 and GEPS5 forecasts and how much these compromise the results? One way to assess what differences are related to the initialization could be to show figures of the day 1 difference, since notable differences in the SSTs/ice at this point should be dominated by the different initialization. All this needs to be discussed in the revised paper.
- 4. You emphasise the importance of the MJO and the sensitivity of MJO forecasts to coupling, but no comment is made about MJO forecasts in GEPS6 versus GEPS5. Has this been looked at previously? How does/might this impact your interpretation of MJO dependent impacts?

**MINOR COMMENTS**

Section 2.3: I don't follow the reasoning here. You say that you can't directly compare the coupled and uncoupled forecasts because the start dates differ, and so you rather compare the two biases instead. However, unless I misunderstood something about the exact computation, this ends up being the same thing: (GEPS6-ERA5)-(GEPS5-ERA5)=GEPS6-GEPS5. So your bias difference plots are just showing GEPS6-GEPS5 anyway. I don't think you can sidestep the problem that the initialization days are different. You just need to mention this as a confounder and discuss how much you think the results depend on it.

"To test the significance, the degrees of freedom are counted by making the following two assumptions: (a) output from different start times or different ensembles is independent" I guess you mean different ensemble members, not different ensembles. As for the first point, this should be fine as long as the start times are relatively spaced out. Can you comment on the typical distance between start dates? The information is in the supplementary tables but it is convenient if you just state this here for the reader.

Figures 2/3: Can you make the continents more visible? Coastlines blend in with contour lines, making it hard to distinguish the two.

L140: You should include a few lines on how the shift in the Gulf Stream is very likely related to the inability of NEMO at ¼ degree to resolve the Gulf Stream properly, and cite some references for the role of model resolution. I don't know as much about the Kuroshio current, but I'm sure model biases in this current, and likely origins of such biases, have been looked at in past studies, so it would be good to discuss these briefly as well. Alternatively, you could add this discussion to your section 4, but if so, please mention here that you will discuss these biases further in section 4.

L150: The link between the Aleutian and Icelandic lows is known and documented, see e.g. Honda et al. (2001): <a href="https://doi.org/10.1175/1520-0442(2001)014<1029:ISBTAA>2.0.CO;2">https://doi.org/10.1175/1520-0442(2001)014<1029:ISBTAA>2.0.CO;2</a> Please add some references here.

L171/172: "indicating that the coupling results in a colder SST" Can you add a comment on why this might be? This comment might be related to the above comment about past literature on Kuroshio current biases in models.

L208: Figure 3d-f should presumably refer to Figure 1d-f.

L211: "Future numerical studies are needed to gain a deeper understanding." Figure 1f and 1i show an NAO pattern in the Euro-Atlantic. The relationship between changes in the Gulf Stream and changes in the NAO have been investigated in many past studies, see e.g. this paper and references therein: <a href="https://doi.org/10.1029/2025GL117228">https://doi.org/10.1029/2025GL117228</a>
More pragmatically, the NAO is the dominant mode of variability there so if you change the SSTs in this region then the atmospheric change is very likely going to project onto the NAO. Please add some comments on this, especially on the past literature.

L216: "We also notice a possible teleconnection from the Aleutian Low through the Arctic into Icelandic Low via a Rossby wave train." Since this teleconnection is known (see above), you should rephrase to rather say that the changes to the Aleutian Low affect the Icelandic Low via a Rossby wave train, and then cite Honda again.

L233/234: "Second, there is a need for more physical understanding of how two-way coupling produces better air—sea fluxes." There are some classic relevant studies on this. Most notably, Barusgli and Battisti (1998) needs to be mentioned here: <a href="https://doi.org/10.1175/1520-0469(1998)055<0477:TBEOAO>2.0.CO;2">https://doi.org/10.1175/1520-0469(1998)055<0477:TBEOAO>2.0.CO;2</a>
In this paper they clearly explain the effect of coupling versus no coupling on heat flux and surface temperature variability. In particular, the low frequency variability in surface

temperature (and, I believe, heat fluxes) will be wrong in uncoupled models due to the excess thermal damping effect they explain there. This is fundamentally related to the fact that the ocean acts as an infinite sink/source of energy in an uncoupled simulation. It seems plausible that changes in the latent heat flux bias variance you see could be related to this. You don't necessarily need to demonstrate this decisively, but some comments at least are necessary.